# Participatory Action Research for Tackling Distress and Burnout in Young Medical Researchers: Normative Beliefs before and during the Greek Financial Crisis

**DOI:** 10.3390/ijerph191710467

**Published:** 2022-08-23

**Authors:** Dimitra Sifaki-Pistolla, Enkeleint A. Mechili, Evangelos Melidoniotis, Alexandros Argyriadis, Evridiki Patelarou, Vasiliki-Eirini Chatzea

**Affiliations:** 1Clinic of Social and Family Medicine, School of Medicine, University of Crete, 71003 Heraklion, Crete, Greece; 2School of Health Sciences, Frederick University, Nicosia 3080, Cyprus; 3Department of Healthcare, Faculty of Health, University of Vlora, 9401 Vlora, Albania; 4Department of Surgery, University Hospital of Heraklion, 75100 Heraklion, Crete, Greece; 5Department of Nursing, School of Health Sciences, Hellenic Mediterranean University, 71004 Heraklion, Crete, Greece; 6Hellenic Mediterranean University, 71410 Heraklion, Crete, Greece

**Keywords:** participatory action research, professional burnout, psychological stress, research personnel, economic recession, normative beliefs, qualitative research

## Abstract

(1) Background: We aimed to explore Young medical researchers (YMR) normative beliefs and perceived causes of distress and burnout, prior and during the financial crisis in Greece, and to assess their views on Participatory Action Research (PAR) interventions towards tackling these disorders. (2) Methods: A Participatory Learning and Action (PLA) methodology was performed in two time periods (prior crisis: December 2008; during crisis: February–March 2017). In both time periods, three different groups (Group 1: females, Group 2: males, Group 3: mixed) of 5–7 participants and two sessions (≈1 h/session) per group took place in each site. Overall, 204 sessions with 1036 YMR were include in the study. (3) Results: Several socio-demographic characteristics of YMR altered during the crisis (lower income, higher smoking/alcohol consumption, etc.). The majority of YMR conceived distress and burnout as serious syndromes requiring professional support. Feeling very susceptible and the necessity for establishing PAR interventions were frequently reported during the crisis. Numerous (a) barriers and (b) cues to action were mentioned: (a) lack of time, money and support from friends/family/colleagues (b) being extensively informed about the intervention, participation of their collaborators, and raising awareness events. (4) Conclusions: The changing pattern of Greek YMR’s beliefs and needs during the crisis stresses the necessity of interventions to tackle distress and burnout. Effectiveness of these interventions could be enhanced by the suggested cues to action that emerged from this study.

## 1. Introduction

Young researchers encounter numerous challenges, both academically and professionally, including establishing expertise in their discipline, balancing personal and work life and developing work opportunities [1]. Nevertheless, there is limited evidence in the literature as regards the prevalence of mental disorders, such as distress and burnout, among young medical researchers (YMR) [1,2]. The aforementioned studies have highlighted the crucial need of further studying distress and burnout in YMR due to its increasing burden, especially during the European financial crisis. Thus, studies should focus on what is utterly missing from the literature and assess these outcomes multifacetedly. Capturing normative beliefs and perceived causes of distress and burnout in this population group is considered a core step towards this direction since it will facilitate a more comprehensive assessment of their mechanisms in order to inform effective tackling measures. 

Professional researchers are among the most vulnerable professions for developing distress and burnout due to the nature of their work [3,4]. Therefore, it is important to identify the individuals at risk or those experiencing difficulties early in their professional trajectory, hear and learn from their “voice” and engage them in targeted interventions [5]. Participatory Action Research (PAR) [6] is among the most reliable approaches to inform and implement effective interventions in the context of mental health. PAR is “a systematic, participatory approach to inquiry that enables people to extend their understanding of problems or issues and to formulate actions directed toward the resolution of those problems or issues” [7]. The research team actively collaborates with the participants in the entire research process [8]. It can be briefly summarized as “social research for social change” [9]. More specifically, it aims to involve ordinary community members in expressing their beliefs and perceptions, and generating practical knowledge about issues and problems of concern through promoting personal and social change [10].

The need of such interventions is imperative since the prevalence of mental problems tend to increase during times of financial recessions or austerity measurements in which related socio-economic stressors are more evident [11,12,13]. This applies for several Northern European countries and especially for Greece, which has been affected more than any other European country from the ongoing financial crisis [14,15]. The impact of austerity in Greece has vastly affected the mental status of the population [16], which has been found to worsen [17]. In particular, distress and burnout in YMR has significantly increased after the onset of the financial crisis, while several socio-demographic predictors have already been identified (marital and family status, income, type of employment, working period, social activity, smoking and alcohol consumption, existence of chronic and other mental health diseases) [2]. Therefore, the exploitation of social-cognitive predictors to enhance intervention enrollment is of high importance for Greece.

The present article focuses on the identification of normative beliefs and perceived causes of distress and burnout in YMR before and during the austerity period in Greece. The main objectives set were: (a) to explore young researchers’ wishes, preferences and perceived benefits of PAR interventions and (b) to identify levers and barriers towards establishing future effective interventions for tackling distress and burnout among professionals in medical research.

## 2. Materials and Methods

### 2.1. Theoretical Framework and Participatory Action Research

The current study was driven by the Normalization Process Theory (NPT) [18,19,20]. The utilization of the NPT framework aimed to investigate levers and barriers towards tackling distress and burnout through the implementation of effective interventions. Within the context of NPT, elements from the Health Belief Model (HBM) [21] were merged in order to investigate the normative beliefs, perceived causes, wishes, preferences and benefits on tackling distress and burnout. 

NPT focuses on the social processes and the individual and collective actions towards implementing an effective intervention. It consists of the four following constructs: understanding (coherence), engagement (cognitive participation), enactment (collective action) and appraisal (reflexive monitoring) [20]. The first two constructs were the main point of focus of the current study. Furthermore, HBM, a psychological health behavior change model, was utilized to explain and predict health-related behaviors. HBM suggests that individuals’ beliefs about health problems, wishes and preferences, perceived benefits of action and barriers to action, as well as self-efficacy and cues to action can explain engagement in health-promoting behaviors [22,23].

Guided by the aforementioned theoretical frameworks, a qualitative case study based on the Participatory Learning and Action (PLA) [24] research method was conducted in two different study periods (Period A: December 2008; Period B: February-March 2017). PLA is a practical, adaptive research strategy that enables diverse groups and individuals of different socio-economic background to learn, work and act together in a cooperative manner. More importantly it enables them to focus on issues of joint concern, express openly their ideas and wishes, identify challenges and generate positive responses in a collaborative and democratic manner [24,25,26,27]. In this study, “PLA-brokered dialogue” technique was utilized to create a level playing field, where all expressed perspectives count, and the knowledge embedded in them is shared and enhanced “around the YMR table” [27]. More details on PLA-brokered dialogue are provided in Section 2.3. 

### 2.2. Setting and Recruitment Processes

Every medical and nursing education department in Greece was involved in this study; both those affiliated to universities (7 medicine and 2 nursing departments) and technical institutions (8 nursing departments). The involved departments that comprised the study sites were located in seven different geographical regions (Athens, Thessaloniki, Sparta, Patra, Ioannina, Larissa/Lamia and Crete). A list of the YMR activating in each department was provided by the “Special Account for Research” from all institutions, while it was randomized in order to identify potential participants from each one of the seven sites. 

Participants were enrolled based on three major inclusion criteria: (a) individuals that had gained their first or second medical/health degree during the past five years; (b) individuals that were employed or occupied voluntarily in the selected sites with research-oriented tasks; and (c) individuals that were less than 40 years old. Young individuals that were assigned only administrative tasks and had a permanent contract with the university were excluded, since this is considered an entirely different type of personnel in the Greek universities. 

All individuals were informed about the purposes and processes of the study and agreed to participate by signing a consent form. 

The main aim of the selected recruitment process was to achieve the maximum variation among the participants in terms of age, gender, medical discipline (i.e., public health, epidemiology, biostatistics, clinical medicine, laboratory research, etc.) and institutional affiliation (university or technical institution). In both time periods, three different groups (Group 1: females, Group 2: males, Group 3: mixed) of 5–7 participants and two sessions (≈1 h/session) per group took place in each site. Overall, 204 sessions of 1036 participants were included in the study (102 sessions and 518 participants per time period).

### 2.3. PLA Sessions and Content Analysis

The twenty-one PLA-brokered dialogues were led by independent and specially trained researchers. One encounter/coordinator and one observer participated in each session, which followed a flexible brainstorming form of discussions and PLA-style interviews [27], aiming to extract responses and capture the voice of the participants, through a collaborative and democratic dialogue. Each coordinator was responsible for the optimum implementation of the PLA techniques and was guiding the conversation by using a predefined topic list. The topic list included questions on: socio-demographic profile of the participants; individual and family health problems and needs (focus on mental health disorders); experiences with mental health services; perceived causes of mental health disorders (focus on distress and burnout-prior and during the financial crisis); normative beliefs on distress and burnout in YMR (prior and during the financial crisis); wishes, preferences and perceived benefits of interventions and services offering mental health support (prior and during the financial crisis); and levers and barriers towards tackling distress and burnout in YMR. Appendix A presents a summary of the questions raised during the PLA sessions, while it describes the theoretical components of NPT and HBM that were measured.

All sessions were audio recorded upon receiving participants’ written approval. Furthermore, PLA charts were developed to ensure that verbal and visual forms of data were recorded in a consistent manner among the diverse PLA groups. PLA charts were formed using colored papers and post-its and were later digitalized in order to preserve and protect data. Extensive filed notes were taken by the PLA observer who was also responsible for the transcription of the audio files, the filling of the fieldwork evaluation forms (coding framework) and the development of a PLA narrative report for each session. 

Principles of thematic analysis in qualitative research were adopted to analyze the collected data using narrative reports [28]. Prior thematic analysis, we transcribed all recordings of the PLA sessions. Then, the analysis was led by DSP in collaboration with the researchers involved in the PLA sessions. A deductive thematic analysis [29] informed by NPT and HBM was performed with the aim to examine the emergent data and integrate them with the theoretical framework’s components (Appendix A). Specifically, we defined the themes based on the theoretical framework’s components and then coded the relevant key-words per theme in order to summarize the results. 

### 2.4. Ethical Standards

The “Helsinki declaration” as well as all the regulations that govern the research activities in Greece were strictly followed in this study. The study was approved by the Board of Trustees of the 2nd Health Region of Piraeus and Aegean Islands (protocol: 652008). All participants were informed about the purposes of this study and agreed to participate and be audio recorded during the PLA sessions by signing individual’s consent forms. 

## 3. Results

### 3.1. Participants’ Profile

The participants’ profile vastly altered during the two time periods. It is indicative that prior to the economic crisis YMR lived alone, had a mean income of 985 euros, a smaller proportion of them were smokers or heavy alcohol consumers and the prevalence of distress and burnout among them was relatively low. On the contrary, during the economic crisis the majority of the participants lived with family members, presented a decreased mean income of 535 euros, many were tobacco or heavy alcohol consumers, while increased distress and burnout prevalence was observed. A detailed description of participants’ socio-demographic profile prior and during the financial crisis is presented in Table 1.

### 3.2. Causes and Normative Beliefs on Distress and Burnout

Several causes and normative beliefs on distress and burnout were mentioned during the conduct of the PLA sessions in both time periods. Table 2 presents the top five causes and the five most frequent normative beliefs reported by the participants in period A and in period B. Briefly, prior to the economic crisis, the participants stressed that experiencing “other mental health diseases” or “traumatic events” are major causes for developing distress. In addition, “academic pressure”, “chronic health issues” and “high expectations of self and others” were also frequently reported. Contrary to that, in period B, “financial issues”, “high expectations of self and others” and “occupational insecurity” occupied the first three positions. Furthermore, “academic pressure” and “family and interpersonal issues” were frequently mentioned in period B. During the economic crisis, most of the normative beliefs were changed; distress was considered a frequent disease, partially serious and hardly self-controlled, with no stigma for others, but stigma for themselves, that causes feelings very susceptible and with serious impacts on everyday life that cannot be managed without professional support. 

Further alterations on the reported causes and normative beliefs were also observed as regards burnout development (Table 2). Working over hours, feeling of lacking control and getting stuck in a rut, economic difficulties and professional insecurity, insufficient rewards and having little time to relax or failing to care yourself were the five major reported causes in period B. Burnout contrary to distress was not considered stigma for others and themselves neither before nor during the financial crisis.

### 3.3. Experiences with Mental Health Services, Wishes, Preferences and Perceived Benefits 

In period A, 14.1% of the PLA participants had previous experience with mental health support services; 60.3% of them were satisfied with the provided services. Only a small percentage (11%) of the participants had experience with participatory interventions, reporting high satisfaction. In period B, more participants reported having experience with mental health support services (37.1%), however a smaller percentage of them (28.1%) were satisfied. In addition, only 4.7% of them had engaged in participatory interventions that let them all highly satisfied (Table 3). 

Most of the participants were not willing to engage in participatory interventions prior the financial crisis, whereas during the crisis, 74.1% of them were positive and willing to mobilize themselves and their colleagues to collectively benefit from the profits gained from participatory interventions. Similar changes were observed in other reported wishes/preferences regarding participatory interventions and mental health support services aiming to tackle distress and burnout in YMR (Table 3). During the financial crisis, 63.7% of the PLA participants were willing to build up a shared understanding of the participatory interventions’ aims, but did not feel ready to sustain and engage themselves (16.7%) or change their risk behaviors (15%). 

Furthermore, all participants could recognize the benefits of participating in such interventions. The three most frequently reported perceived benefits in period A were “feeling less distressed and tired”, “feeling stronger and ready to work effectively” and “feeling relived and cheerful”. In period B, “feeling less distressed, tired and emotionally exhausted”, “feeling being supported by professionals and others” and “forget the economical and work-related anxieties” were most frequently mentioned. 

### 3.4. Levers, Barriers and Cues to Action in Tackling Distress

Levers, barriers and cues to action towards tackling distress and burnout through participatory interventions were discussed in PLA sessions and grouped accordingly in Figure 1. Most of the parameters were similar in both study periods, while some levers, barriers and cues to action were expressed only in period B. These are the following: (a) levers: abetment from the seniors/directors and involvement of role playing and/or active participation of the researchers; (b) barriers: existing bad interpersonal relationships with colleagues, lack of money or accessibility to healthcare services and lack of support from the family/friends; (c) cues: advice and encouragement from the seniors/directors and no charge for the participants in the interventions. In addition, lack of free time and negative attitude from colleagues or seniors/directors were among the most frequent barriers in both periods. Moreover, participation of other colleagues, conduct of these interventions during the working hours, and clearly explaining the expected benefits on mental health and the overall professional efficacy were among the major levers both before and during the financial crisis. 

## 4. Discussion

The importance of preventing adverse mental health outcomes, such as burnout and distress, in work environments has been widely emphasized in the literature [30,31]. The current study managed to capture and map the normative beliefs and perceived causes of distress and burnout in YMR working in the medical and nursing departments in Greece, prior and during the austerity period. The identified wishes, preferences, perceived benefits, levers, barriers and cues to action for participating in interventions and mental health support services stressed the changing patterns that emerged after the economic recession.

Greek YMR living and working in a stressful socio-economic environment conceive distress and burnout as serious syndromes that could be managed with professional support, while they feel very susceptible during the economic crisis. Most of them recognize the need for mental health support through PAR interventions and are willing to enroll upon the approval and support of other colleagues and mainly their seniors. Lack of time, money and support from friends/family and directors were among the main perceived barriers. Lastly, based on the reported cues to action, it is obvious that YMR are willing to participate only upon being extensively informed about the intervention and when their collaborators are also aware and motivated to enroll.

Numerous studies come in accordance with the present findings as regards the factors that can trigger (perceived causes) anxiety or burnout in the workplace. High demands, low perceived control, inadequate social support, effort–reward imbalance and job insecurity contribute to common mental disorders, including anxiety and depression [32,33,34,35]. As supported in the literature [36], harsh socio-economic recession periods tend to deteriorate these factors through the rapid increase of psychosocial stressors in the workplace; reflecting later to an increased burden of work-related stress and burnout [2,36]. Similar to the presented findings, after the onset of the financial crisis, burnout syndrome has been correlated more with high demands, lack of tasks control, extreme workload, low reward, job insecurity and lower adaptive organizational attitudes [37,38]. 

In addition, lack of control, “getting stuck in a rut”, insufficient rewards and failing to care for oneself were mentioned by the study participants only upon the onset of the crisis. Furthermore, before the crisis, distress was related to health-related issues, while during the crisis it was more frequently attributed to financial, occupational and interpersonal-related causes. Other studies verify these observations and further stress the adverse effects of any fiscal crisis [36,39,40]. 

According to the above, three broad categories of work-related factors that explain how work may contribute to the development of distress and burnout have been identified in the literature and verified by the current study: imbalanced job design, occupational uncertainty and lack of value and respect in the workplace [41]. These factors may have contributed to the normative beliefs of the under-study population group. Notably, Greek YMR’s normative beliefs on distress and burnout before and during the austerity period revealed negative attitudes towards these outcomes. The immense psychosocial effects of the crisis were also implied by the Greek YMR. They considered distress and burnout frequent severe diseases in their workplace, significantly burdening their everyday life, while they stress the perceived stigma for distress and doubted whether they could deal with it by themselves. 

The present results conveyed new knowledge on the current normative beliefs and the strong need for improved psychosocial work environments to prevent distress and burnout among YMR. They imply the importance of organizational interventions that is also illustrated by the fact that structural work environment factors are vastly influencing the development of mental disorders; mainly distress and burnout. The fact that the source of burnout often lies in structural work environment factors points at the potential value of multifactorial PAR interventions [38].

### 4.1. Strengths and Limitations

Τo the best of our knowledge, this is the first study worldwide assessing YMR’ views on distress and burnout as well as on participatory interventions. The utilization of PLA approach, a robust research method, is among the major strengths of this study. It offered the ideal democratic environment for discussion and enabled the expression of deeper thoughts and opinions over the under-discussion topics. NPT and HBM added value to this research by providing a strong conceptual theoretical framework for appraising the nature of YMR’ attitudes and coherence. Beliefs, wishes and preferences of participants of different socio-demographic background were equally assessed and interpreted under the NPT and HBM theoretical models. The conduct of the study in multiple geographical sites across the country and the enrollment of all medical and nursing departments contributed to the external validity of the findings. In addition to that, information and classification biases, which could be inserted due to the qualitative nature of this study, were minimized by the utilization of trained researchers that performed the data coding and the interpretation of the results.

Nevertheless, the current results should be discussed under the light of some limitations, including those imported by default in all qualitative studies (inability to capture quantitative indicators, difficult to investigate causality, etc.). Additionally, the slightly diverse demographic profile (gender and age) and academic background (physicians, nurses, biostatisticians, epidemiologists, social workers, etc.) of participants could have impacted their responses. Nevertheless, it was unavoidable due to the fact that the overall researchers’ profile changed in Greece after the onset of the financial crisis. The authors believe that this is not likely to affect the main results substantially, since variation between gender and age distributions were minor and not statistically significant. Furthermore, the current findings could be representative only for the YMR working in the Greek medical or nursing departments, rather than for any institution undertaking biomedical research (e.g., psychology or pharmacology departments). Still, these departments are not fully engaged to biomedical research since they cover a wider scientific spectrum, while at the same time they tend to collaborate with researchers affiliated to medical or nursing departments. Lastly, it should be noted that there is a time gap between data collection and publication due to the large amount of work required to thematically analyze the data for the big sample of YMRs and lastly, cross-validate the accuracy of the components, themes and key findings. 

### 4.2. Implications

Moving one step further from the current findings, PAR interventions in YMR should be embraced and enhanced by a work-environment team in each research department/faculty. This could offer continuous monitoring of the interventions, stimulate researchers and seniors’ interest to engage, promote the intervention and its expected benefits, as well as plan workshops (less lectures, more counseling and participatory activities) and other activities to maintain continuity and integration [23,42]. Individual and team level emotional status should be prioritized throughout the interventions by embedding additional joint activities in daily practice (e.g., sharing work and tasks, boosting mutual support) [25]. Furthermore, all procedures should be standardized and evaluated by following-up mechanisms and the utilization of questionnaires to assess measurable indicators [42].

## 5. Conclusions

The findings of this study could open the discussion on mental disorders among YMRs and invite future researchers to either follow-up current findings or to give more scientific data about this problem in different populations and countries. After that, robust findings could be utilized to establish a comprehensive organizational work-oriented approach towards prevention of distress and burnout in YMR; especially in low-middle resource countries or regions with similar socio-economic background as Greece. Efforts should be made towards minimizing the environmental and psychological job stressors, while at the same time enhancing buffering factors both at work and personal environment. YMR and their seniors should be further informed about the significance and the necessity of PAR interventions for tackling distress and burnout. This could be acquired by focusing on the major levers, barriers, wishes and preferences expressed, as well as on the suggested cues to action that emerged from the present study. Future research should focus on the particularities observed in European countries with diverse profile during the ongoing financial crisis. 

## Figures and Tables

**Figure 1 ijerph-19-10467-f001:**
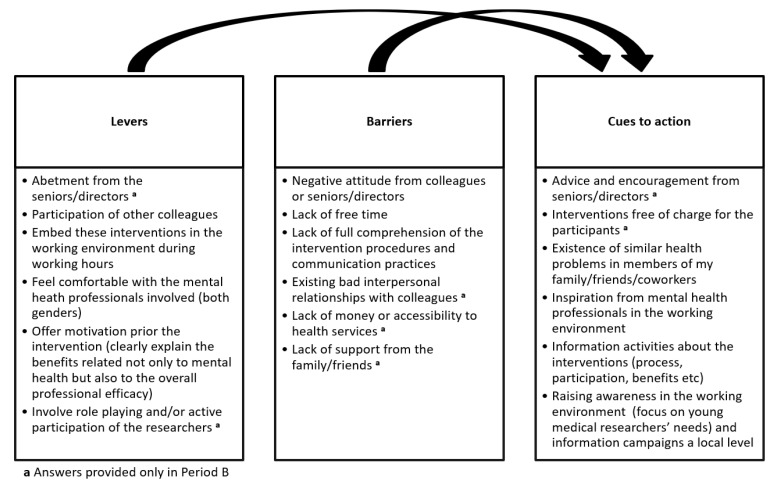
Levers, barriers and cues to action in tackling distress and burnout through participatory interventions before and during the financial crisis.

**Table 1 ijerph-19-10467-t001:** Participants’ socio-demographic profile (n = 1036).

Characteristics	Period A	Period B
	N (%)	N (%)
**Gender**		
*Males*	261 (50.4)	253 (48.8)
*Females*	257 (49.6)	265 (51.2)
**Age** * (years)	32 (3)	34 (4)
**Marital status**		
*Single*	261 (50.4)	284 (54.8)
*Divorced/Widower*	65 (12.5)	138 (26.6)
*Married/Engaged*	192 (37.1)	96 (18.5)
**Family members** * (individuals)	2.5 (1)	1.5 (0.5)
**Children** * (individuals)	0.5 (0.5)	2 (0.5)
**Siblings** * (individuals)	1 (0.5)	0.5 (0.5)
**Household status**		
*Living alone*	238 (35.5)	134 (25.9)
*Living with husband/wife/partner*	184 (35.5)	111 (21.4)
*Living with family*	81 (15.6)	176 (34.0)
*Living with other type of roommate*	15 (2.9)	97 (18.7)
**Type of employment**		
*Permanent*	142 (27.4)	38 (7.3)
*Free lancer*	326 (62.9)	380 (73.4)
*Volunteer*	50 (9.7)	100 (19.3)
**Income *** (Euro)	985 (160)	535 (90)
**Working period *** (years)	4.5 (1.5)	2.5 (1.5)
**Type of research unit**		
University	265 (51.2)	280 (54.1)
Technical Institution	253 (48.8)	238 (45.9)
**Social activity** * (days/week)	4 (0.5)	1 (1)
**Ever smokers**	172 (33.2)	273 (52.7)
**Heavy alcohol consumers** (≥4 cups/week)	119 (23.0)	170 (32.8)
**Existence of chronic diseases *** (number)	0.5 (0.5)	3.5 (1)
**Existence of mental health diseases *** (number)	0.5 (0.5)	1.5 (1)
**Distress diagnosis**	54 (10.4)	196 (37.8)
**Burnout diagnosis**	112 (21.6)	303 (58.5)

* Median; IQR.

**Table 2 ijerph-19-10467-t002:** Causes and normative beliefs on distress and burnout before (n = 518) and during (n = 518) the financial crisis.

Distress in Period A		Distress in Period B	
Major Causes	Normative Beliefs	Major Causes	Normative Beliefs
Other mental health issues (95%)	Not frequent disease (78%)	Financial issues (98.5%)	Frequent disease (82.5%)
Traumatic events (94.5%)	Not serious/can be controlled easily (77%)	High expectations of self and others (85.8%)	Partially serious/can be controlled hardly (82%)
Academic pressure (93.5%)	No stigma for others/perceived stigma for themselves (68.5%)	Occupational insecurity (85.5%)	No stigma for others/perceived stigma for themselves (75.6%)
Chronic health issues (70%)	Not feeling easily susceptible (65%)	Academic pressure (80.2%)	Feeling very susceptible (75.4%)
High expectations of self and others (69%)	Serious impact on everyday life/could be managed by oneself (63.2%)	Family and interpersonal issues (76.4%)	Very serious impact on everyday life/could not be managed by oneself alone (70.9%)
**Burnout in Period A**		**Burnout in Period B**	
**Major Causes**	**Normative Beliefs**	**Major Causes**	**Normative Beliefs**
Work over hours (62.7%)	Not frequent syndrome (77.5%)	Work over hours (86.4%)	Frequent disease (92%)
Academic pressure (61.5%)	Not serious/can be controlled easily (75.3%)	Lack of control and getting stuck in a rut (86%)	Very serious/can be controlled hardly (79.8%)
Losing sight of your expectations (59%)	No stigma for others/nor perceived stigma for themselves (62.8%)	Economical barriers and professional insecurity (85.9%)	No stigma for others/nor perceived stigma for themselves (72.1%)
Little time to relax and failing to care oneself (58%)	Not feeling easily susceptible (56.2%)	Insufficient rewards (83.2%)	Feeling very susceptible (72%)
Bad interpersonal relationships with colleagues (56.5%)	Serious impact on everyday life/could be managed by oneself (51.8%)	Little time to relax and failing to care oneself (82.7%)	Very serious impact on everyday life/could be managed by oneself (69.5%)

All answers are accredited by the majority of the participants (n > 50%). Top five causes are reported, following ascending order.

**Table 3 ijerph-19-10467-t003:** Experiences, wishes/preferences and perceived benefits on mental health support services and participatory interventions before (n = 518) and during (n = 518) the financial crisis.

Period A		
Experiences	Wishes/Preferences	Perceived Benefits ^a^
Having experience (n = 73; 14.1%)	Willing to participate to participatory intervention (n = 196; 38.4%)	Feeling less distressed and tired
Satisfying experience (n = 44; 60.3%)	Willing to organize myself and colleagues to collectively contribute to the work involved in participatory interventions (n = 196; 38.4%)	Feeling stronger and ready to work effectively
Experience with participatory interventions (n = 8; 11%)	Jointly with my colleagues build up a shared understanding of the participatory intervention aims (n = 184; 36.1%)	Feeling relived and cheerful
Satisfying experience with participatory interventions (n = 8; 100%)	I am ready to sustain and continuously behave in line with a behavioral change intervention (n = 182; 35.7%)	Bond with my colleagues
	Ready to change risk behaviors (n = 160; 31.4%)	Remember the important things in life (apart from work)
**Period B**		
**Experiences**	**Wishes/Preferences**	**Perceived Benefits ^a^**
Having experience (n = 192; 37.1%)	Willing to participate to participatory intervention (n = 378; 74.1%)	Feeling less distressed, tired and emotionally exhausted
Satisfying experience (n = 5; 28.1%)	Willing to organize myself and colleagues to collectively contribute to the work involved in participatory interventions (n = 378; 74.1%)	Feeling being supported by professionals and others
Experience with participatory interventions (n = 9; 4.7%)	Jointly with my colleagues build up a shared understanding of the participatory intervention aims (n = 325; 63.7%)	Forget the economical and work-related anxieties
Satisfying experience with participatory interventions (n = 9; 100%)	I am ready to sustain and continuously behave in line with a behavioral change intervention (n = 85; 16.7%)	Remember the important things in life (apart from work)
	Ready to change risk behaviors (n = 76; 15%)	Spend more value time to myself

^a^ Top five perceived benefits are reported, following ascending order.

## Data Availability

Not applicable.

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
