# Peer review of "Participatory Action Research for Tackling Distress and Burnout in Young Medical Researchers: Normative Beliefs before and during the Greek Financial Crisis"

_ijerph, 2022, doi:10.3390/ijerph191710467_

Round 1
Reviewer 1 Report
Very interesting topic for this nice manuscript. Burnout in Medical REsearchers during the famous Greek Crisis.
The paper is well written and in a cognitive way, and it is definitely of interest for the readers of the journal.
Introduction sets the scene very well.
I have some concerns on the Methods. REsearchers with administrative duties were excluded? This is speculative. In fact, the main flaw of the paper is the fact that it does not manage to rest objective even in some parts of the design. PLA figure should be remove in the some context. Good in a PPT presentation, but not here.
I think the paper should be accepted, but the authors need to do a better job, in maintaining the data presentation as objective as in Table 1, rather than Table 2 for example.
This is a very relevant area, but need to keep it simple and objective.
Author Response
Very interesting topic for this nice manuscript. Burnout in Medical REsearchers during the famous Greek Crisis. The paper is well written and in a cognitive way, and it is definitely of interest for the readers of the journal. Introduction sets the scene very well.
Response: Thank you very much for your comments and for reviewing the manuscript. We have revised it according to your comments below.
I have some concerns on the Methods. REsearchers with administrative duties were excluded? This is speculative. In fact, the main flaw of the paper is the fact that it does not manage to rest objective even in some parts of the design. PLA figure should be remove in the some context. Good in a PPT presentation, but not here.
Response: This study aimed to assess the situation among researchers not administrative staff. In the Greek Universities permanent administrative staff is under a totally different type of employment (permanent contract, salary etc) and has only administrative and bureaucratic tasks, not research oriented. At the same time, we have researchers who may deal with research and scientific tasks alone, or combine administrative and research tasks. Both types of researchers were included. The only case we might have excluded individuals, is when they were permanent administrative personnel of the university with no research tasks. Towards this direction we have revised the methods and clarified who we excluded and why.
(See lines 230-233: Young individuals that were affiliated as “researchers” but had been assigned only administrative tasks and had a permanent contract with the university were excluded, since this is considered an entirely different type of personnel in the Greek universities.)
Additionally, as regards PLA figure we omitted it from the main manuscript and added it in the supplementary material. We truly want to publish it even as a suppl. Material because most of the PLA studies use it for credibility purposes. Thank you once again.
I think the paper should be accepted, but the authors need to do a better job, in maintaining the data presentation as objective as in Table 1, rather than Table 2 for example.
Response: Table 1 is a descriptive table for the participants profile characteristics. Table 2 summarizes their statements and is a qualitative analysis table, therefore it could not fully alter. However, with respect to your comment we have now added percentages per cell. (Lines 241-244)
This is a very relevant area, but need to keep it simple and objective.
Response: Thanks for the remark. We revised the manuscript as described above.

Reviewer 2 Report
The article made a very good impression.
The Introduction provide sufficient information and have relevant references. Method and Results are very well presented. Discussion is in accordance with the Results.
Author could only improve the conclusions. They sad that the study could utilized to establish a comprehensive organizational work-oriented approach. I think at first that this study could invite future researcher, preferably following-up study, to give more scientific data about this problem and than the results in future could be utilized to establish same preventive procedures.
Author Response
The article made a very good impression.
The Introduction provide sufficient information and have relevant references. Method and Results are very well presented. Discussion is in accordance with the Results.
Response: Thank you very much for your comments and for reviewing the manuscript. We have revised it according to your comments below.
Author could only improve the conclusions. They sad that the study could utilized to establish a comprehensive organizational work-oriented approach. I think at first that this study could invite future researcher, preferably following-up study, to give more scientific data about this problem and than the results in future could be utilized to establish same preventive procedures.
Response: Thank you for your remark. We have revised the conlsusions according to your comment. Specifically, we noted that
“The findings of this study could open the discussion on mental disorders among YMRs and invite future researchers to either follow-up current findings or to give more scientific data about this problem in different populations and countries. After that, robust findings could be utilized to could be utilized to establish a comprehensive organizational work-oriented approach towards prevention of distress and burnout in YMR; especially in low-middle resource countries or regions with similar socio-economic background as Greece.” (See Lines 410-413)

Reviewer 3 Report
The work presented is of enormous interest and relevance. It is worth paying attention to the levels of distress and exhaustion that various health collectives, in this case researchers, present in order to be able to take measures to solve these situations.
Below are a series of recommendations to improve the quality of the work presented:
- In the abstract should appear briefly the methodology followed in the research.
- Similarly, in the methodology it is not clear what the "dialogue negotiated by PLA" is.
- The authors should include more evidence about the PLA methodology.
-Likewise, the data analysis process followed should be explained in more detail.
- Although the results and analysis are adequate, it is surprising that the authors are presenting this work in 2022 when the fieldwork data are from 2017. This should be justified.
Author Response
The work presented is of enormous interest and relevance. It is worth paying attention to the levels of distress and exhaustion that various health collectives, in this case researchers, present in order to be able to take measures to solve these situations.
Response: Thank you very much for your comments and for reviewing the manuscript. We have revised it according to your comments below.
Below are a series of recommendations to improve the quality of the work presented:
- In the abstract should appear briefly the methodology followed in the research.
Response: We have revised the methods section, as follows “Methods: Participatory Learning and Action (PLA) methodology was conducted performed in two time periods (prior crisis: December 2008; during crisis: February-March 2017). In both time periods, three different groups (Group 1: females, Group 2: males, Group 3: mixed) of 5-7 participants and two sessions (≈1hour/session) per group took place in each site. Overall, 204 sessions with 1,036 YMR were include in the study. (See Lines 21-27)”
- Similarly, in the methodology it is not clear what the "dialogue negotiated by PLA" is.
- The authors should include more evidence about the PLA methodology.
-Likewise, the data analysis process followed should be explained in more detail.
Response: We have revised this section according to your comment. We’ve added details per section keeping in mind the words limit for original research papers (See methods section).
- Although the results and analysis are adequate, it is surprising that the authors are presenting this work in 2022 when the fieldwork data are from 2017. This should be justified.
Response: Indeed, there is a major time gap between the fieldwork data and the publication date. The main reason for this has to do with the large amount of data and qualitative analysis time demands. We had to work for one full year to finalize data analysis and cross-validation of the final findings, since we ‘ve noticed minor errors in the first place and decided to re-check it again. As you understand qualitative studies are usually using smaller samples for these reasons. But we attempted to conduct a nationwide study and therefore had large sample and very long recordings and transcription files, etc. We’ve added the following sentences in Lines 391-395: Lastly, it should be noted that there is a time gap between data collection and publication due to the large amount of work required to thematically analyze the data for the big sample of YMRs and lastly, cross-validate the accuracy of the components, themes and key findings.

Round 2
Reviewer 1 Report
Good modifications. It can be accepted now.
Reviewer 3 Report
The changes made by the authors comply with the requirements made by this reviewer.